# The *DACH1* Gene Transcriptional Activation and Protein Degradation Mediated by Transactivator Tas of Prototype Foamy Virus

**DOI:** 10.3390/v15091899

**Published:** 2023-09-08

**Authors:** Yongping Ma, Jie Wei, Jing Song, Zhongxiang Hu, Ruifen Zhang, Zhi Li, Yan Sun

**Affiliations:** 1College of Life Science, Shaanxi Normal University, Xi’an 710119, China; 2College of Biological Science and Engineering, North Minzu University, Yinchuan 750021, China; 3College of Environment and Life Sciences, Weinan Normal University, Weinan 714099, China

**Keywords:** DACH1, PFV, Tas, PPM1E, SUMOylation, degradation

## Abstract

Foamy viruses are members of the *Retroviridae* family’s *Spumaretrovirinae* subfamily. They induce cell vacuolation and exhibit a foamy pathogenic impact after infecting cells. DACH1 (dachshund family transcription factor 1) is a crucial cytokine linked to tumor development, and is associated with the growth of many different malignant tumor cells. Additionally, DACH1 suppresses pancreatic cell proliferation and is involved in diabetes insulin signaling. Prototype foamy viruses (PFVs) were used for the investigation of the regulatory mechanism of FVs on cellular DACH1 expression. The results show that DACH1 expression in PFV-infected cells was inconsistent at both the transcriptional and protein levels. At the transcriptional level, *DACH1* was significantly activated by PFV transactivator Tas, and dual-luciferase reporter gene tests, EMSA, and ChIP assays found a Tas response element of 21 nucleotides in the *DACH1* promoter. PFV and Tas did not boost the levels of DACH1 protein in a manner consistent with the high levels of DACH1 transcription expression. It was noted that Tas increased the expression of the Ser/Thr protein phosphatase PPM1E, causing PPM1E-mediated post-translational SUMOylation alterations of DACH1 to prompt DACH1 to degrade. The reason for DACH1 protein degradation is that DACH1 inhibits PFV replication. To sum up, these findings show that PFV upregulated the transcription of DACH1, while urging its protein into PPM1E-mediated SUMOylation, to eliminate the adverse effect of DACH1 overexpression of host cells on viral replication and promote virus survival.

## 1. Introduction

Foamy viruses, which are members of the *Retroviridae* family and belong to the *Spumaretrovirinae* subfamily, set themselves apart by the paradox of their pathogenicity [1]. PFV infection differs significantly across cells and animals, causing a severe cytopathic effect in adherent fibroblast cultures but no identifiable correlation with illness in living organisms [2,3]. The PFV genome is approximately 11–12 kb, making it the longest genome among the RNA viruses in the family of retroviruses that have been discovered to date [4]. In the cDNA of PFV, there are two long terminal repeat (LTR) sequences at the 5′ and 3′ ends, respectively. Downstream of the 5′ LTR sequence are the structural protein genes *gag*, *pol*, and *env*, while upstream of the 3′ LTR sequence are the important regulatory protein genes encoding transactivator Tas and Bet [5,6,7]. After the successful invasion of PFV into the host cell, due to the low basal activity of the internal promoter (IP) and the involvement of certain cytokines, a small amount of Tas protein can be translated. IP has a higher affinity for Tas protein compared to the LTR promoter region. Once these small amounts of Tas protein continuously bind to IP and transcribe and translate more Tas protein, the expression of Tas protein reaches a certain threshold. Tas protein then binds to the LTR promoter region to initiate the transcription and expression of the three structural proteins Gag, Pol, and Env, and then new viral particles assemble [8,9].

As an important transcription factor of PFV, a nonpathogenic retrovirus, Tas plays a crucial role in viral replication by binding to the Tas response element (TRE), which is located in the LTR and IP, initiating the transcription of PFV [10]. Tas is a crucial target of numerous canonical cell factors affecting host immunity, and is crucial for viral lifecycle functions. Both p300 and PCAF can acetylate Tas, and its overexpression can increase the activity of the PFV LTR promoter. The acetylation of Tas increases its affinity for the DNA binding domain and improves its capacity to bind to viral promoters [11,12]. PML (promyelocytic leukemia) can interact with the N-terminal region of PFV Tas through its zinc finger domain to form complexes, which can inhibit PFV transcription and play a key role in the antiviral mechanism of IFN [13]. It was discovered that Nmi (N-Myc interaction factor) interacts with Tas and isolates it in the cytoplasm, limiting the transactivation of Tas dependent on PFV LTR and IP and inhibiting PFV replication [14].

The *DACH* gene encodes two putative transcription factors (DACH1 and DACH2) and shares high conservation among *Homo sapiens* and *Drosophila melanogaster*. The human dachshund gene is featured by two domains: DachBox-N and DachBox-C. DachBox-N of DACH, known as the DACH Ski/Sno (DS) domain due to the approximately 35% conservation to the Ski/Sno family of co-repressors, possesses a crystal structure of a winged helix forkhead and interacts with some classical nuclear factors, such as NCoR, HDAC3, and Six6, to regulate cell proliferation and differentiation [15]. DachBox-C, the carboxy-terminal structural domain of the *DACH1* gene, is associated with the ubiquitin-binding enzyme Ubc9 [16].

DACH1 is an essential retinal determination gene network (RDGN) member that dramatically impacts metazoan development, regulating ocular, limb, brain, and gonadal development [17,18]. However, a growing body of literature indicates that DACH1 has a more significant role in mammalian cells. NMP9 is downregulated by DACH1 to prevent breast cancer cell invasion and metastasis [19]. Furthermore, it inhibits lung adenocarcinoma invasion and tumor growth by repressing CXCL5 [20]. DACH1 protein abundance is inversely correlated with the expression of cyclin D1 in human renal cancers and endometrial cancer [21,22]. DACH1 is broadly involved in cancer tumorigenesis, cell proliferation, and invasion via different pathways. One prominent characteristic of the *DACH1* gene is that it has two CpG islands, where DNA 5-cytosine methylation with 5′-CG-3′ dinucleotides, found in the promoter region, regulates epigenetic modification and impacts downstream gene expression [23,24]. DACH1 directly binds p53 with its carboxyl terminus and enhances p53-dependent cell cycle arrest [25]. In another study, DACH1 repressed cyclin D1 transcription through association with the AP-1 protein to suppress cellular proliferation, S phase progression, and clone formation [21]. The expression of DACH1 was found to interact with smad4 to inhibit TGF-β-induced apoptosis [26,27]; conversely, Dach1 depletion was more susceptible to apoptosis in alveolar epithelium cells due to the modulation of C-Jun/Bim activity [28].

However, the role of DACH1 in the response of a host infected by a virus is under-documented. Here, we present our findings from a comprehensive investigation of the effect of Tas on regulating *DACH1* gene expression. A major discovery was that Tas could bind directly to the promoter region of DACH1 and boost its transcription level while promoting protein breakdown via PPM1E-dependent SUMOylation.

## 2. Materials and Methods

### 2.1. Cell Culture, Plasmids, and Transfection

HeLa, HT-1080, and 293T cell lines were cultivated in Dulbecco’s modified Eagle medium (DMEM) supplemented with 1% penicillin, streptomycin, and 10% fetal calf serum. The plasmid pKW-DACH1 was a gift from Professor Kongming Wu (Huazhong University of Science and Technology, Wuhan, China), which containing the ORF sequence of human *DACH1* gene, and pcDNA3.1-DACH1 was subcloned from pKW-DACH1. Plasmid pHSRV13, which can express the PFV provirus DNA, was provided by our laboratory. His-SUMO1 was subcloned from pOTB7-SUMO1(MiaoLing Biology, Wuhan, China) into a mammalian expression vector pcDNA4/myc-His B. Overexpression plasmids of proteins encoded by PFV, pCI-Gag, pCI-Pol, pCI-Env, pCI-Bet, pCI-Tas, and pCMV-Myc-Tas were constructed previously in our laboratory. Plasmid transfection was performed by Lipofectamine 2000 (Thermo Fisher, Waltham, MA, USA) following the manufacturer’s protocol. pGL3-Basic, pRL-TK, and pGL3-control were purchased from Promega (Shanghai, China).

### 2.2. PFV Collection, Quantification, and Infection

PFV stocks were generated and amplified by HT-1080 cells by transfecting infectious pHSRV13. After HT-1080 cells were transfected with pHSRV13, all PFV elements were produced and assembled to form new viral particles. The medium containing viral particles was collected and concentrated by using an Amicon Ultra-15 centrifugal filter (Millipore, Burlington, MA, USA) and kept at −80 °C.

The indicator cell line BHK-21-pAc-5′LTR-EGFP, which contains a plasmid encoding the enhanced green fluorescent protein (EGFP) driven by the PFV LTR, was used to measure viral titer. The PFV Tas protein can activate the LTR promoter, resulting in the expression of EGFP. Therefore, the viral titer was assessed by the indicator cells with EGFP fluorescence; the procedure has been described previously [29].

HeLa cells were plated in a 6-well plate at a density of 5 × 10^5^ cells the day before infection. The next day, the culture medium was removed, and cells were treated with a diluted PFV stock containing polybrene (final concentration 8 g/mL) for 4 h at 37 °C. Then, the cells were washed three times with 0.01 M phosphate-buffered saline (PBS, pH 7.2) and cultured continuously in 10% FBS.

### 2.3. Quantitative PCR

The mRNA levels of DACH1, PPM1E, and GAPDH were assessed by real-time quantitative PCR (RT-qPCR) on a BioRad CFX96 Real-Time System. After HeLa cells were transiently transfected by Tas or infected by PFV, the total RNA was isolated using RNAiso (Takara, Dalian, China), and the RNA was reverse transcribed to first-strand cDNA by M-MLV (RNase H-) reverse transcriptase (Takara). The primers were as follows: 5′-CCTTGACAAACTCTCTCTAACTGG-3′ (sense) and 5′-CTTGAGCTCTGGCATTATC.

TATGG-3′ (antisense) for DACH1, 5′-GGTGCACCAAAGAAAGCAAA-3′ (sense) and 5′-CTCCCCTGTTGAACCCAAAT-3′ (antisense) for PPM1E, and 5′-GAAGGTGAAGGTCGG.

AGTC-3′ (sense) and 5′-GAAGATGGTGATGGGATTTC-3′ (antisense) for GAPDH as an internal control. Gene expression data were obtained using TransStart Tip Green qPCR SuperMix (TransGen Biotech, Beijing, China), the relative amount of mRNA compared to the internal control was calculated using the 2^−ΔΔCT^ method [30], and GAPDH was used as the endogenous control.

### 2.4. Dual-Luciferase Reporter Assay

The plasmid pUC57-DACH1 was purchased from the company GENWIZ (Suzhou, China), which containing the promoter region fragment of *DACH1* gene ranging from −2000 to 200 bp with the translation initiation site as +1. PROME (https://alggen.lsi.upc.es/cgi-bin/promo_v3/promo/promoinit.cgi?dirDB=TF_8.3 (accessed on 4 April 2021)) and JASPAR (https://jaspar.genereg.net/ (accessed on 4 April 2021)) online software were used to analyze the transcription factor binding sites and compare them with the Tas response site of the internal promoter (IP) of PFV and the TRE of the *CDKN1C* sequence, which was confirmed. A series of reporter plasmids were constructed into a pGL3-Basic vector (Promega, Madison, WI, USA) containing parts of DACH1 promoter regions by PCR-mediated amplification from pUC57-DACH1; the primers are listed in Appendix A. A series of mutant vectors was created to further confirm the Tas direct-acting site of the DACH1 promoter sequence. The pGL3-DACH1 plasmid served as a template, and the primers containing the mutant sites are listed in Appendix A.HeLa cells were seeded into a 48-well culture plate at a density of 5 × 10^4^ cells/well and transfected with pGL3-DACH1 (300 ng) or promoter sequence deletion plasmids pGL3-dS1 to pGL3-dS8 (300 ng), respectively, or promoter site mutation plasmids pGL3-mS1 to pGL3-mS5 (300 ng), respectively, with pCI-Tas (30 ng), pGL3-Basic (300 ng) and pCI-Tas (30 ng) as a negative control, and pGL3-control (300 ng) and pCI-Tas (30 ng) as a positive control. pRL-TK (3 ng) was co-transfected into each well to normalize the transfection efficiency, and Lipofectamine 2000 was used for plasmid transfection. After transfection for 48 h, the firefly and Renilla luciferase activities were detected following the manufacturer’s instruction of the Dual-Luciferase Reporter Assay System (Promega). The results are shown as the ratio of firefly luciferase activity to Renilla luciferase activity.

### 2.5. Electrophoretic Mobility Gel Shift Assay (EMSA)

EMSA experiments were performed with biotin-labeled DNA probes using the LightShift Chemiluminescent EMSA Kit (Thermo Fisher). The probes used for EMSA assays included unlabeled DACH1-TRE and 5′-end oligonucleotide biotin-labeled DACH1-TRE, which was 5′-CTGGTTGGATAATTTGGGTTA-3′, and the mutant probe DACH1-mTRE was 5′-AGTTGGTGATAATTTGGGTTA-3′. Nuclear extracts were prepared from 293T cells transfected with a plasmid expressing Tas protein with MYC-tag. For competition experiments, a mutant probe was added in 100-fold excess; the unlabeled probe was added in 5-fold, 25-fold, and 100-fold molar excess during the preincubation period; and 1.5 mg MYC-tag monoclonal antibody (Proteintech, Wuhan, China) was added for supershift experiments.

### 2.6. Chromatin Immunoprecipitation Assay

Chromatin immunoprecipitation (ChIP) was performed using the ChIP Assay Kit (Beyotime Biotechnology, Shanghai, China). Briefly, HeLa cells were plated at a density of 2.5 × 10^6^ cells in 10 cm dishes, 16 μg pCI-neo or pCI-Tas plasmids, respectively, transiently transfected for 48 h, and then crosslinked with formaldehyde. Chromatin was sonicated, incubated, and precipitated with Tas antiserum or normal rabbit IgG (negative control), respectively. The immunoprecipitated DNA fragments were detected by PCR and RT-qPCR analysis. The primers for the DACH1 promoter (−692 to −529 bp) were 5′-CGTCCCTCCCA-CAGTTTCTT-3′ (sense) and 5′-AATACCCTTTGAAGCCGCAA-3′ (antisense).

### 2.7. RNA Interference

For the RNAi tests, the RIBOBIO Corporation (Guangzhou, China) supplied three specific siRNA duplexes and a negative control targeting distinct locations of PPM1E and UBE2I. The HeLa cells were plated on 12-well cell culture plates and incubated until confluence reached approximately 70%. Following the manufacturer’s recommendations, siRNA and pCI-Tas were co-transfected using the Lipofectamine LTX transfection reagent (Thermo Fisher). The final concentration of siRNA in each well was 50 nM. After 48 h of cell incubation, RNAi effects were detected by conducting Western blotting on the extracted protein.

### 2.8. SUMOylation Site-Directed Mutation Assays

According to the DACH1 siRNA sequence given in the literature [18,31], the designed and synthesized DACH1 shRNA sequence (Appendix A) was constructed into the shRNA lentiviral expression vector pGreenPuro (System Biosciences, Palo Alto, CA, USA) according to the instructional process, co-transfected with pSBH155 plasmid and pSBH156 plasmid into 293T cells, and packaged into lentiviruses. The lentiviruses containing a final concentration of 8 μg/mL of polybrene were injected into HeLa cells. After 48 h of infection, DACH1 knockdown cells were obtained by screening with the addition of a final concentration of 1 μg/mL puromycin, and this was named the HeLa-DACH1.KD cell line.

The software GPS-SUMO was used to estimate the potential alteration sites of DACH1 by SUMOylation. Based on the three predicted sites (K348, K599, K631), primers for SUMOylation modification site-directed mutations (Appendix A) were designed following the manufacturer’s instructions for the Fast MultiSite Mutagenesis System Kit (TransGen Biotech) to generate a mutant of DACH1 with lysine to arginine substitutions at these three sites. Using the previously created pcDNA3.1-DACH1 expression vector as a template, the mutant fragments were amplified and assembled according to the kit’s instructions, resulting in the expression vector pcDNA3.1-DACH1-3mS with mutations in the three SUMOylation modification sites of the DACH1. HeLa-DACH1.KD cells were plated at a density of 2 × 10^5^ cells in a 12-well plate, pcDNA3.1-DACH1 (500 ng) or pcDNA3.1-DACH1-3mS (500 ng) were co-transfected with the pCI-Tas (500 ng) plasmid by Lipofectamine2000. After 48 h of transfection, protein samples were collected and quantified, and western blotting detected protein expression.

### 2.9. Co-Immunoprecipitation

HeLa cells were plated at a density of 3 × 10^6^ in a 10 cm dish and transfected with pCI-neo or pCI-Tas with His-SUMO1 for 48 h; then, cells were washed with ice-cold PBS containing 20 mM N-ethylmaleimide and lyzed with lysis buffer (Beyotime Biotechnology). Anti-normal rabbit IgG or anti-His (Proteintech) was added to the tube and rotated at 4 °C overnight. Then, protein A/G magnetic beads (20 μL) were added to the tube and incubated at room temperature for 1 h. The beads were removed from the solution of immune complexes using a magnetic stand and washed three times with the same lysis buffer, and then eluted with SDS-PAGE loading buffer and boiled at 95 °C for 5 min. The extract was subjected to immunoblot analysis using anti-DACH1 (Proteintech) and anti-SUMO1 (ZENBIO, Chengdu, China).

### 2.10. Western Blot Analysis

Protein was extracted from the cell lines with a RIPA lysis buffer (150 mM NaCl, 0.5% sodium deoxycholate, 0.1% SDS, 1 mM EDTA, 50 mM Tris-HCl, pH 7.4, 1 mM PMSF) and quantified using the BCA Kit (Thermo Fisher). Samples of 15 μg of protein were loaded per well in SDS-PAGE and then transferred onto PVDF membranes (Bio-Rad, Hercules, CA, USA). Nonspecific binding of the membranes was blocked in 5% fat-free milk and then incubated separately with diluted primary antibodies overnight at 4 °C. The primary antibodies used were as follows: DACH1 (ab77234, Abcam), PPM1E (ab137122, Abcam), and UBE2I (ab33044, Abcam). GAPDH (60004-1-Ig, Proteintech) was used as a loading control. After washing, the membranes were incubated with horseradish peroxidase-conjugated IgG antibodies for 1 h at room temperature. The protein signals of the PVDF membranes were visualized using an enhanced chemiluminescence detection system (Tanon Corp, Shanghai, China).

### 2.11. Statistical Analysis

All data are presented as the mean ± SD. The two-tailed test was utilized for normally distributed data to compare two groups. A *p*-value of less than <0.05 was considered statistically significant. All analyses were carried out utilizing SPSS 22.0 (Chicago, USA). Graphs were created using GraphPad Prism version 5.0 software (GraphPad Software, USA).

## 3. Results

### 3.1. PFV and Tas Can Activate the Expression of DACH1 Transcription Level

The RT-qPCR assay confirmed the mRNA expression upregulation of DACH1 by PFV and Tas protein. The results show that *DACH1* mRNA expression was increased significantly whether via PFV infection or the Tas protein expression in HeLa cells. But other coded proteins of PFV, such as Gag, Env, Pol, and Bet had no contribution to the transcriptional activation of DACH1 (Figure 1). The results reveal that *DACH1* gene transcriptional activation by PFV was only caused through its transactivator Tas.

### 3.2. Tas Directly Regulates DACH1 Gene Transcription through a Binding Site in the DACH1 Promoter Region

The Tas protein can bind to 5′-LTR and the IP unique site of PFV to start structuring PFV genes and other accessory protein-encoding genes [32,33]. In addition, the p57^kip2^ gene was also shown to be precisely and powerfully activated by the Tas protein of PFV [34]. As *DACH1* mRNA expression was activated by Tas, it was presumed that there was a particular binding site in the DACH1 promoter region for the Tas protein.

Previous studies have revealed PFV IP-TRE [35] and Kip2-TRE sequences [34]; so, compared with PFV IP-TRE and Kip2-TRE binding sites, and simultaneously using transcription factor binding site prediction softwareJASPAR and PROMO (Appendix A), we anticipated critical transcription factor binding sites and designed a variety of reporter plasmids of DACH1 to the firefly luciferase report gene vector pGL3-Basic (Figure 2A). The results show that when the constructed DACH1 promoter region (−2000~200 bp) vector pGL3-DACH1 was co-transfected with the Tas expression vector pCI-Tas, there was a significant difference in relative luciferase activity compared to the transfected empty vector pCI-neo group, indicating that Tas could bind to the promoter region to activate DACH1 transcription. In the constructed promoter sequence deletion vector co-transfection with pCI-Tas, there was no significant difference in relative luciferase activity between pGL3-dS1 and pGL3-dS4 compared to pGL3-DACH1, which shows that there were no Tas response sites on these deletion sequences, and the elements contained in the deletion sequences do not affect the transcriptional activation of the DACH1 by Tas. Starting with the pGL3-dS5 to pGL3-dS8 groups, there was a significant difference in relative luciferase activity compared to pGL3-DACH1, while there was no significant difference between the groups, suggesting that the possible Tas response element is located at the −627~−474 fragment, where a predicted sequence with high similarity to the PFV IP-TRE exists. Based on the above experimental results, we mutated the bases of the predicted sites on the −627~−474 fragment and the base sequence of its adjacent region and constructed five DACH1 promoter site mutation vectors (Figure 2B). The results show that the relative luciferase activities of mutation report vectors pGL3-mD1 and pGL3-mD5 were significantly different from those of pGL3-DACH1 reporting vectors, while there was no significant difference between the other mutation reporting vectors and pGL3-DACH1 reporting vectors. Based on the above luciferase reporter gene experiment, we preliminarily identify the binding site where Tas may act on the DACH1 promoter as 5′-CTGGTTGGATAATTTGGGTTA-3′.

EMSA and ChIP were conducted to validate further that the known DACH1-TRE sequence could be active and form a complex with Tas proteins. As shown in Figure 2C, the binding shift band of Tas was determined to be selective, and the DNA–protein complex was notably eliminated in the presence of 25-fold and 100-fold competition oligonucleotides and 100-fold unlabeled mutant probe loss of binding ability. Additionally, after adding specific antibodies to the Tas and DACH1-TRE complexes, supershift bands were observed. According to the EMSA result, Tas can bind to DACH1-TRE extracellularly.

A ChIP experiment was conducted to confirm the intracellular binding ability. HeLa cells were transfected with pCI-Tas plasmids to determine their true state through PCR and RT-qPCR, while pCI-neo was transfected as a control group. Initially, prepared primers were utilized to amplify 164 bp segments containing the DACH1-TRE by PCR (Figure 2D). The PCR results demonstrate that in the input group, whether pCI-neo or pCI-Tas were transfected, the target band could be detected, while with IgG immunoprecipitation as the negative control, there was no obvious target band. In the Tas antibody immunoprecipitation product, compared with the pCI-neo transfected control group, obvious target bands can be observed. Then, a more sensitive approach, RT-qPCR, was used to detect the combination of Tas proteins and DACH1 promoters (Figure 2E). In the negative control group, low expression levels of DACH1 were detected, and there was no significant difference, while in the ChIP group, the transfected Tas expression vector showed extremely significant differences compared to the transfected empty vector, indicating that the Tas protein can bind to DACH1-TRE. These facts suggest that Tas could bind precisely to the DACH1-TRE of the DACH1 promoter. The sequence similarity and base conservation of previously known Kip2-TRE, IP-TRE, and identified DACH1-TRE were analyzed. The results show that there is no substantial sequence similarity between the three TREs (Figure 3A), but DACH1-TRE has a continuous seven base similarity with IP-TRE (Figure 3B). IP is an internal promoter of PFV and has a high affinity for Tas. This could explain why Tas can bind to and activate DACH1. The base conservation result (Figure 3C) revealed that numerous conserved G bases may have a significant impact on the binding of Tas.

### 3.3. The Protein Expression of DACH1 Was Not Influenced by PFV or Tas

Although Tas could directly combine with the promoter region of DACH1 and increase its accumulation at the transcription level, a contentious issue was also discovered at the protein level of DACH1. Since elevated levels of *DACH1* mRNA were identified in HeLa cells with PFV infection, the protein expression of DCH1 by PFV was investigated further (Figure 4A). As evidenced by the findings, PFV had no discernible effect on DACH1 protein expression. This finding demonstrates that various stages of early events could not influence the levels of the DACH1 protein during PFV infection. Since PFV functioned on host cells and influenced the protein expression only in the whole virus, it was crucial to determine whether the various components of PFV, particularly Tas, might affect DACH1 protein expression. However, neither structural nor nonstructural PFV proteins affected DACH1 (Figure 4B). The results suggest that DACH1 protein expression was unaffected by the presence or absence of Tas compared to WT and pCI-neo (Figure 4C). To further verify the results, experiments were conducted with different cell lines, including HT-1080 and U251 cell lines (results not shown), and the results were consistent with those derived from HeLa. Additionally, as two downstream genes of DACH1 have previously been reporte [23,31,36], the expression levels of cyclin D1 mRNA and p21 protein were examined after pCI-Tas transfection, and the results reveal no significant alterations in either cyclin D1 mRNA expression or p21 protein expression (Appendix A). All these results lead to the compelling conclusion that neither Tas nor PFV influence the protein expression of DACH1.

Therefore, we conclude that despite the complexity of the PFV infection process, the upregulation of the DACH1 transcription level by both PFV infection and Tas overexpression could not give rise to the DACH1 protein. These results suggest that DACH1 may undergo post-transcriptional, translational, or post-translational regulation, resulting in no change in protein expression.

### 3.4. Tas Promoted DACH1 in SUMO-Mediated Protease Degradation Depending on PPM1E

The protease inhibitor MG-132 was used to determine whether DACH1 protein degradation occurred through the ubiquitin–protease pathway. It was discovered that the expression level of DACH1 protein was higher with the treatment of MG-132 under the condition of stable Tas expression (Figure 5A). Meanwhile, without Tas expression, no matter whether this was with or without MG-132, the DACH1 protein level was unchanged. Moreover, DACH1 could be marked by a sharp increase when we used siRNA to knock down UBE2I, a critical E2 ubiquitin-like protein ligase mediating SUMO attachment to the substrate and modifying proteins mostly into regulation or protease degradation (Figure 5B). Immunoprecipitation and Western blot analysis were utilized to detect the SUMOylation of endogenous DACH1 by Tas. In Tas transfected with His-SUMO1 HeLa cells, the expression level of SUMO-DACH1, in which the DACH1 protein is SUMOylated, was significantly higher than in controls transfected with pCMV-Myc. The results demonstrate that Tas could increase the DACH1 SUMOylation level (Figure 5C). To further clarify whether the occurrence of SUMOylation of DACH1 was necessary for its final degradation, three DACH1 knockdown cell lines were constructed using shRNA technology, and the knockdown effect was detected using Western blotting. Compared with the control, the DACH1 expression levels were decreased in all three knockdown cell lines (Figure 5D), and the best knockdown cell line by DACH1-shRNA02, which was named DACH1.KD, was used for subsequent experiments. When we transfected the wild-type DACH1 expression vector pcDNA3.1-DACH1 or DACH1 SUMOylation site mutation expression vector pcDNA3.1-DACH1-3mS, co-expressing Tas protein in the HeLa-DACH1.KD cell line, the DACH1 protein expression of site mutation of SUMOylation was considerably higher than in the wild-type group. The results indicate that the degradation of the DACH1 protein is significantly related to its translational SUMOylation modification (Figure 5E).

A study based on insulin resistance reported that CaMKII phosphorylates and blocks nuclear translocation of hepatocyte HDAC4 under conditions of obesity: lower nuclear HDAC4 decreases the SUMOylation and degradation of the co-repressor DACH1, and finally, causes defective insulin signaling [37]. Through previous microarray assay analysis, it was found that for another gene, named *PPM1E*, expression was upregulated; this gene belongs to the PPM family, and is highly specified to the process of dephosphorylating multifunctional CaMK to regulate their activities negatively [38,39,40]. To expound the aforementioned paradox of Tas on DACH1, the mRNA and protein expression of PPM1E was detected (Figure 6A,B). As is clearly shown, overexpression of Tas could activate the transcription and translation of PPM1E, which resulted in the accumulation of PPM1E in HeLa cells. Meanwhile, after knocking down the expression of PPM1E by siRNA, a dramatic increase in DACH1 was detected (Figure 6C). The above results show that Tas could increase PPM1E expression and lead to the degradation of DACH1.

### 3.5. DACH1 Overexpression Inhibits PFV Replication

Previous research results have shown the inhibitory effect of DACH1 in the cell cycle [18,41], and combining these findings with our results of inconsistencies between *DACH1* mRNA and protein expression, we suggest that the expression of DACH1 may be detrimental to the survival and replication of the virus. Therefore, the virus regulates the degradation of the DACH1 protein to facilitate its replication. To verify this hypothesis, we investigated the impact of PFV structural protein Gag and regulatory protein Tas on PFV replication in PFV-infected cells by detecting differences in mRNA and protein expression levels after DACH1 overexpression. To investigate whether DACH1 also affects viral transcription, RT-qPCR was performed (the primers for the detection of *gag* and *tas* are referenced in the literature [42]), and the results are shown in Figure 7A. In the case of DACH1 overexpression, the mRNA levels of *gag* and *tas* genes significantly decreased compared to the control. Western blot results show (Figure 7B) that in PFV-infected HeLa cells, overexpression of DACH1 resulted in an evident decrease in Gag and Tas proteins compared to the control, indicating that overexpression of DACH1 inhibited PFV gene expression. Meanwhile, we used the indicator cell line BHK-21-pGL4.17-5′LTR-Luc under the same experimental treatment to assess the viral titer of extracellular and intracellular conditions (Figure 7C). The results demonstrate that progeny viruses released from extracellular or intracellular infected production were inhibited in overexpressed DACH1 cells. These results demonstrate that DACH1 may have an influence on PFV replication in the early stages of transcription.

## 4. Discussion

Whether the weak disease association of FV results from the balance between hijacking various cell cytokines to escape persecution by the host immune system and producing more progeny viruses to promote virus survival is still unclear. Curiosity urged us to explore the roles of Tas and some cell cytokines during PFV infections. Our study found the highly expressed *DACH1* gene at the transcriptional level because Tas regulates the *DACH1* gene by directly binding to the promoter region. Both LTR and IP can be efficiently activated by Tas; however, in persistent infection, IP can be efficiently activated at a more robust transcriptional level than the LTR promoter, which cannot be efficiently activated [35,43]. Kip2-TRE has been identified in the human *CDKN1C* gene. Kip2-TRE binds to Tas to form a complex that the FV IP-TRE can efficiently block [34]. This shows that the affinity of Tas for TREs of different promoter regions is variable, but among the Tas response element sequences, one to two conserved guanines are essential for binding to Tas. The mutations in the conserved G bases of Kip2-TRE can partially prevent the binding of Tas and Kip2-TRE. Five Tas response elements have been identified in the PFV LTR U3 region, which requires upstream DNA sequence elements for Tas-dependent transcriptional activation. The other four TRE elements can function independently in a Tas-dependent manner. However, there is no significant sequence similarity among these five Tas response elements [35]. In comparison with DACH1-TRE sequences, Kip2-TRE and IP-TRE did not show significant sequence similarity, but the results of the transcription factor binding site conserved analysis indicate that several vital G bases are essential for Tas binding. This variability may suggest that different cytokines may be required when Tas acts on different target genes, acting in concert with bound Tas, resulting in TRE sequence variability.

In eukaryotic cells, one of the major pathways leading to protein degradation is the ubiquitin–proteasome pathway, and the other is autophagy [44,45]. Our results show that the degradation of the DACH1 protein is due to post-translational SUMOylation modification, which in turn undergoes degradation via the proteasome pathway. SUMO modification is the dynamic and reversible post-translational modification of the protein. Many viral proteins have been demonstrated to undergo SUMO modification, which is essential for viral replication because it enhances the synthesis and assembly of viral macromolecules and suppresses the host immune response [46,47,48,49,50]. In the process of viral infection and replication, viruses can also improve their survival regulation by regulating the level of SUMO in host cells. Viruses, including HIV-1, Zika virus, and coronavirus, have evolved to utilize the host SUMOylation system to counteract the antiviral activity of SUMO proteins and modify their proteins to achieve virus persistence and pathogenesis [51]. The SUMOylation modification of HIV-1 integrase can regulate the affinity of co-factors and facilitate HIV-1 replication [52]; however, the interaction between the p6 domain of the HIV-1 Gag protein and SUMO1 molecule can block its ubiquitination, which is detrimental to HIV-1 replication [53]. The diversity of SUMO functions promotes and inhibits viral infection in different ways [47,48]. However, in our study, the SUMOylation of DACH1 was prepared for further ubiquitination to facilitate the degradation of overexpressed DACH1.

In insulin resistance induced by obesity in mice, it was found that obesity can activate CaMK II, which induced HDAC4 phosphorylation and prevented its nuclear translocation, resulting in a decrease in DACH1 expression level. The decrease in DACH1 protein expression was triggered by the SUMO-mediated modification of the DACH1 protein, accompanied by ubiquitination degradation [37]. Our study confirms that Tas could upregulate the PPM1E gene and protein. Another study on PPM1E conducted in our laboratory also proved that the expression of PPM1E would inhibit the phosphorylation level of HDAC4 and promote the nuclear translocation of HDAC4 [54]. Based on the above-mentioned reports and our findings, we established an explanation for how PFV Tas regulates DACH1 transcription and protein degradation during PFV infection. As shown in Figure 8, Tas initially binds to the Tas response element in the DACH1 promoter, leading to the transcription of the *DACH1* gene. This is due to the fact that the DACH1 promoter contains sequences highly similar to the PFV IP-TRE. However, the expression of DACH1 suppresses PFV replication, which is detrimental to PFV survival. As a result, the virus enhances the expression of PPM1E through Tas. Overexpression of PPM1E inhibits the phosphorylation level of CaMK II, leading to a decrease in the phosphorylation level of HDAC4, promoting its nuclear translocation, and, correspondingly, promoting the SUMOylation of DACH1 protein, ultimately degrading it via the ubiquitination pathway.

What is puzzling is the transcriptional activation of DACH1 by the viral transactivator Tas and Tas-mediated ubiquitination modification and degradation of the DACH1 protein. During long-term evolution, viruses have acquired the ability to disrupt cytokines that have adverse effects on their replication, thereby facilitating the development of infections, such as TRIM5 and APOBEC3G found in retroviruses [55]. Similar research has reported that infection of bone marrow-derived dendritic cells (BMDCs) and porcine alveolar macrophages (PAMs) with porcine reproductive and respiratory syndrome virus (PRRSV) of different genotypes and strains can inhibit the expression level of SLA-DR mRNA. However, PRRSV deubiquitinates the SLA-DR protein through its NSP2 ovarian tumor domain, promoting the expression of total SLA-DR protein and functional SLA-DR protein on the cell surface. The inconsistent expression between protein and mRNA levels indicates that SLA-DR plays a novel role in the immune response after PRRSV infection [56]. DACH1-TRE was identified in the promoter region of the *DACH1* gene, which can be recognized by the viral TAS protein. Also, it was predicted that several cellular transcription factors, such as AIRE, ElK-1, POU2F2, VDR, NFI/CTF, HNF-1B, Cart-1, POU3F2, R2, and RUNX1, might bind to DACH1-TRE. Previous studies have shown that in acute myeloid leukemia with RUNX1 mutations, *DACH1* mRNA expression is upregulated after RUNX1 mutations occur [57]. In myeloid cells, C/EBP α and GATA-1 can directly bind to the promoter of DACH1 and act as its transcription inhibitor, having a role in cell proliferation [58]. Thus, in the PFV-infected cell, the *DACH1* gene might be regulated by both Tas and other cellular factors, which might result in the inconsistent expression pattern of DACH1 at transcription and translation levels. Our research results indicate that DACH1 inhibits PFV replication, which may be the reason PFV ultimately degrades DACH1 through Tas, thereby facilitating the development of persistent infection by the virus in cells. These results are informative for subsequent studies on the function of DACH1, which may be prepared as an antiviral target. Moreover, the findings of this study may have implications for other retroviruses, such as HIV, HTLV, etc.

## 5. Conclusions

In summary, we investigated the molecular pathways involved in regulating DACH1 transcriptional activation and protein degradation by PFV, particularly PFV Tas, and identified DACH1 as a novel inhibitor of PFV replication. The SUMOylation of the DACH1 protein and its destruction by PPM1E are the result of the virus exploiting the SUMO modification pathway in order to manage the unfavorable intracellular environment of the host and enhance its survival.

## Figures and Tables

**Figure 1 viruses-15-01899-f001:**
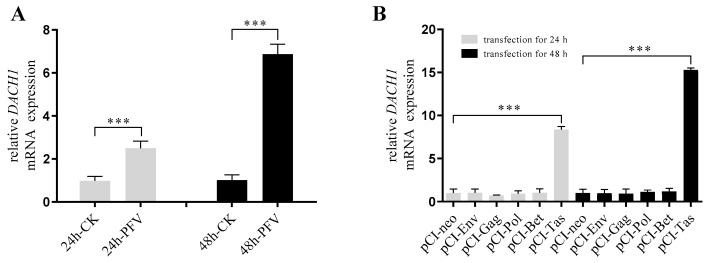
PFV promotes the expression of DACH1 mRNA dependent on its transactivator Tas. (**A**) DACH1 mRNA expression was facilitated by PFV infection. After HeLa cells were infected by PFV for 24 h or 48 h at MOI = 0.2, RT−qPCR was performed to detect the mRNA expression of DACH1. The normal cultured HeLa cells served as a control group (CK group). (**B**) DACH1 mRNA expression increase was dependent on the Tas of PFV. HeLa cells transfected with the empty vector pCI−neo plasmid after 24 h and 48 h were used as the control group and compared with the viral protein expression plasmids. All data are presented as mean ± SD (*n* = 3). *** *p* < 0.001.

**Figure 2 viruses-15-01899-f002:**
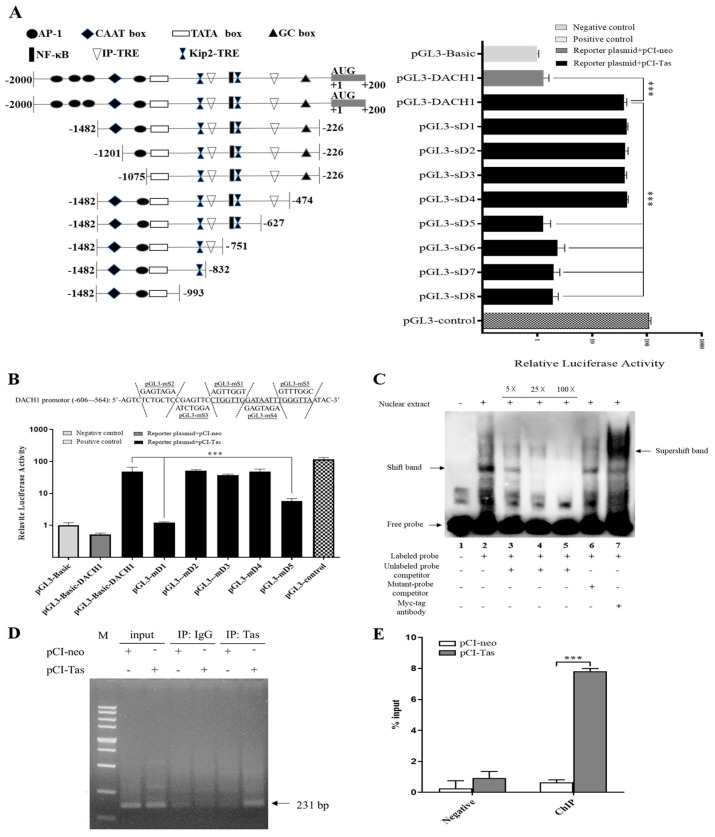
Presence of Tas response elements in the DACH1 promoter region. (**A**) Tas binding fragment in the promoter of DACH1 using DACH1 promoter sequence deletion reporters were detected by dual-luciferase report assay. The promoter fragment activity was represented by firefly luciferase and Renilla luciferase ratios, with pGL3−Basic reporter activity as a negative control and pGL3−control reporter activity as a positive control. (**B**) DACH1−TRE was identified using DACH1 sequence mutation promoter reporters by dual-luciferase report assay. Mutant vector construction based on pGL3−DACH1 and partial sequences, including mutation sites of vectors, are separated by dotted lines, as shown. (**C**) EMSA confirms that Tas and DACH1−TRE can bind specifically in the extracellular space. The labeled probe was incubated with 8 μg of nuclear extract for 20 min, and unlabeled DACH1−TRE (lanes 3 to 5) or mutant DACH1−TRE (lanes 6) oligonucleotides were used as competitors for another 20 min. After the labeled probe and nuclear extract were incubated, the antibody was added to continue for 1 h, and the formed nuclear protein and DACH1−TRE complex were supershifted to a position marked by an arrow (lane 7). (**D**) Verification of intracellular interactions between Tas protein and DACH1−TRE by ChIP−PCR. PCR amplifies the adjacent sequences of DACH1−TRE, using input samples as the positive control, normal IgG immunoprecipitation samples as the antibody-specific control, and the pCI−neo transfection group immunoprecipitated with the Tas antibody as the transfection negative control. (**E**) Intracellular interaction of the Tas protein and DACH1−TRE was validated by ChIP−qPCR. RT−qPCR was utilized to detect the DACH1 promoter fragment of the Tas antibody immunoprecipitation product. Normal Ig−G immunoprecipitation transfected pCI−neo and pCI−Tas groups were used as the negative control; Tas antibody immunoprecipitation transfected pCI−neo and pCI−Tas groups were used as ChIP experimental groups. All data are presented as mean ± SD (*n* = 3). *** *p* < 0.001.

**Figure 3 viruses-15-01899-f003:**
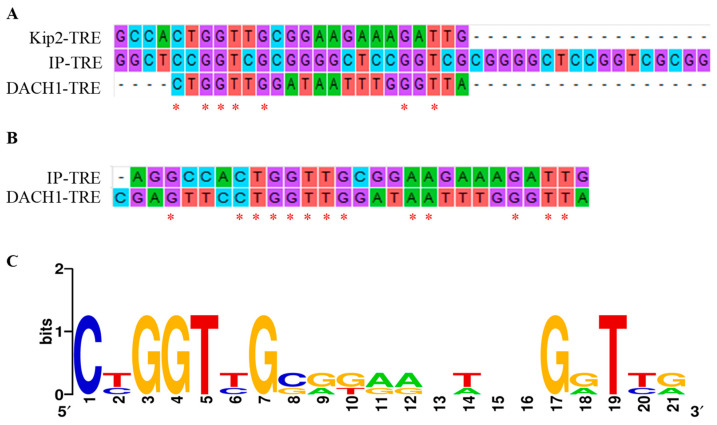
Analyses of sequence similarity and base conservativeness in various TREs. (**A**) Sequence similarities of Kip2−RE, IP−TRE, and DACH1-TRE using ClustalW. (**B**) Sequence similarities of IP-TRE and DACH1-TRE using ClustalW. (**C**) Analysis of the base conservation of TREs by Weblogo.

**Figure 4 viruses-15-01899-f004:**
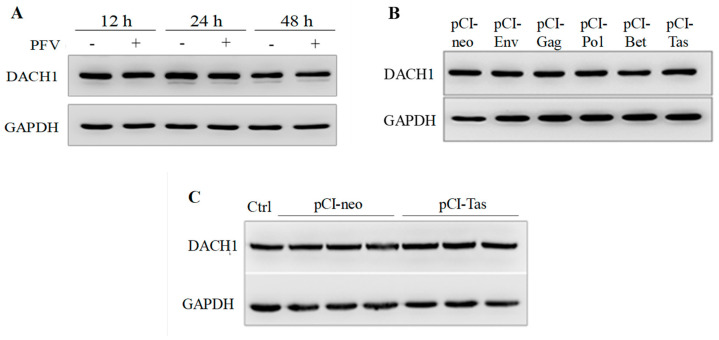
No significant difference in DACH1 protein expression in either PFV−infected or Tas-transfected HeLa cells. (**A**) No significant difference in DACH1 protein expression in PFV−infected HeLa cells at MOI = 0.2 after different infection times. (**B**) DACH1 protein expression was unchanged and transfected with PFV encoding protein. Five proteins encoded by the PFV genome were expressed in HeLa cells for 48 h to detect the expression of DACH1, using untreated HeLa cells as a control. (**C**) Tas cannot affect the expression of the DACH1 protein. After pCI−neo or pCI−Tas was transfected into HeLa cells for 48 h, Western blot analysis was used to test the expression of DACH1.

**Figure 5 viruses-15-01899-f005:**
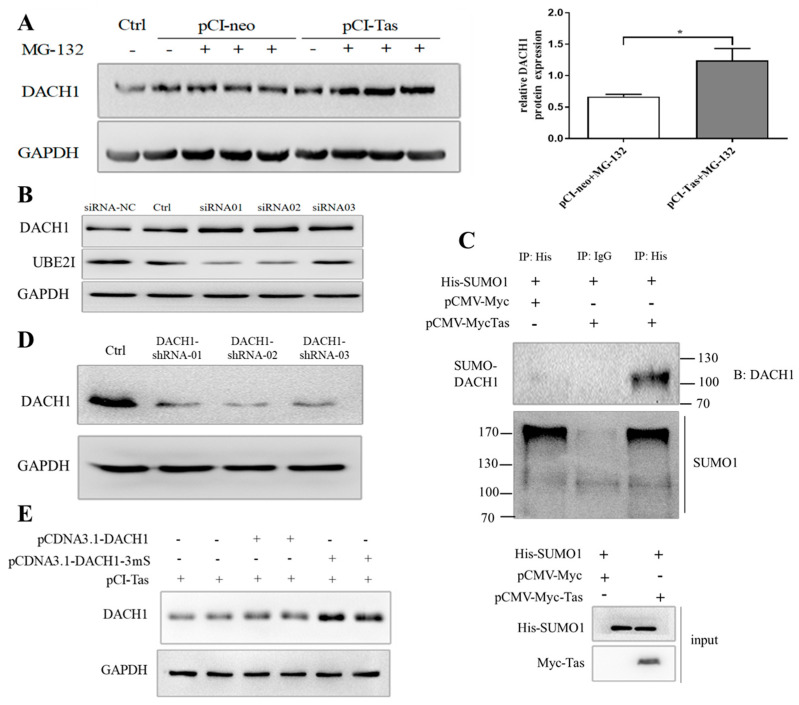
The degradation of DACH1 proteins via the SUMO−mediated pathway. (**A**) DACH1 protein degradation inhibition by protease inhibitor MG−132. After pCI−neo or pCI−Tas transfection for 24 h, MG−132 (40 μM) was added 1 h after treatment; then, we collected proteins for Western blot detection of DACH1 protein expression. (**B**) DACH1 protein degradation was inhibited by UBE2I RNAi. The collected proteins from HeLa cells were co-transfected with UBE2I siRNA (50 nM) and pCI-Tas for 48 h to detect UBE2I and DACH1 protein expression. siRNA−NC was used as the negative control and untreated HeLa cells were used as the control for the normal experimental group. (**C**) Tas promotes DACH1 SUMOylation. Lysates from HeLa cells transfected with pCMV−Myc or pCMV−Myc−Tas were immunoprecipitated using anti−His and blotted with anti−SUMO1 or DACH1. The input blot shows SUMO1 and Tas in whole HeLa cell lysates transfected with pCMV−Myc or pCMV−Myc−Tas. (**D**) Knockdown of DACH1 in HeLa cells by shRNA. Western blotting was used to detect the endogenous protein expression of DACH1 by shRNA lentivirus-infected HeLa cells, with untreated HeLa cells as the control. (**E**) SUMOylation of DACH1 proteins is essential for the degradation of DACH1 by Tas. In DACH1 knockdown of the cell line HeLa−DACH1.KD, the wild-type DACH1 expression plasmid pcDNA3.1-DACH1 or DACH1 SUMOylation site mutation expression plasmid was transfected with pCI−Tas for 48 h, and Western blotting was used to detect the expression of DACH1. All data are presented as mean ± SD (*n* = 3). * *p* < 0.05.

**Figure 6 viruses-15-01899-f006:**
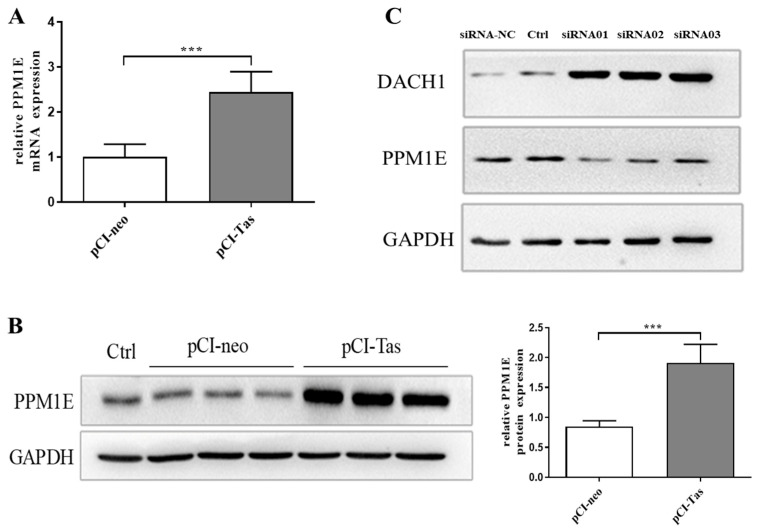
Tas affects DACH1 expression level by regulating PPM1E expression. PPM1E mRNA (**A**) and proteins (**B**) were enhanced by Tas. After transfection of pCI−neo or pCI−Tas for 48 h, cells were collected for RT−qPCR or Western blotting to detect DACH1 expression. (**C**) Tas enhances DACH1 protein expression after interfering with PPM1E via RNAi. PPM1E siRNA (50 nM) was transfected into HeLa cells with pCI−Tas, with siRNA−NC as the siRNA transfection negative control, and untreated Hela cells as the control for the normal experimental group. All data are presented as mean ± SD (*n* = 3). *** *p* < 0.001.

**Figure 7 viruses-15-01899-f007:**
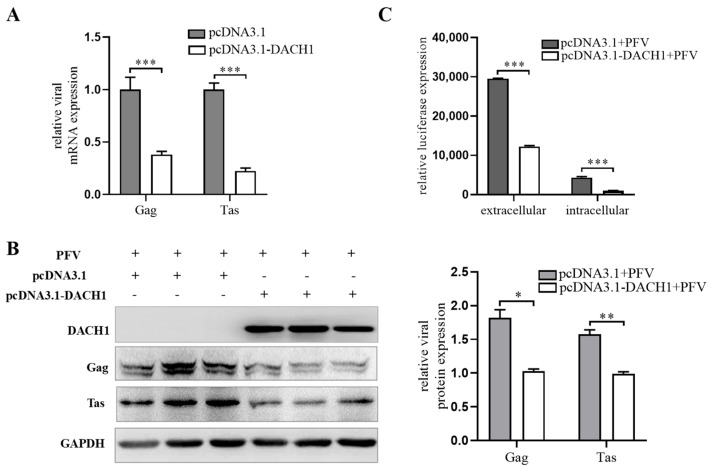
DACH1 overexpression inhibits PFV replication. (**A**) DACH1 inhibits *gag* and *tas* mRNA expression. HeLa cells were infected by PFV at MOI = 0.2 for 24 h in a 6−well plate, and then cells were transfected with pcDNA3.1 (3 μg) or pcDNA3.1−DACH1 (3 μg) for another 48 h. RT−qPCR was used to analyze the mRNA expression of viral *gag* and *tas*. (**B**) DACH1 inhibits Gag and Tas protein expression. After the same treatment as (**A**), Western blot analysis was conducted to determine the effect of DACH1 overexpression on the expression of viral protein Gag and Tas. (**C**) After the same procedure as (**A**), culture medium supernatant was collected for the detection of extracellular viral titer in the medium using indicator cell line BHK−21−pGL4.17−5′LTR−Luc. The adherent cells were digested with trypsin and then 1 mL of fresh culture medium was added. Freeze–thaw cycles were conducted 3 times in liquid nitrogen, after which cells were maintained at 37 °C in a water bath, in which they released intracellular viral particles into the culture medium for detection of the intracellular viral titers. All data are presented as mean ± SD (*n* = 3). * *p* < 0.05. ** *p* < 0.01. *** *p* < 0.001.

**Figure 8 viruses-15-01899-f008:**
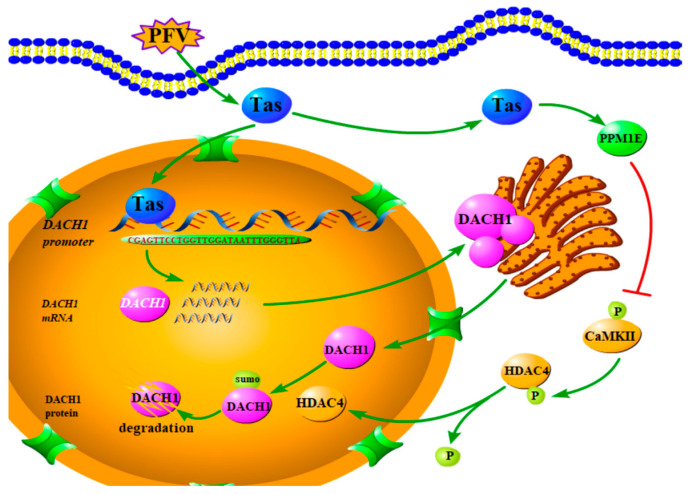
The pathway of DACH1 transcriptional activation and protein degradation mediated by PFV Tas.

## Data Availability

All data supporting reported results can be found in this publication and the Appendix A.

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
