# Peer review of "The DACH1 Gene Transcriptional Activation and Protein Degradation Mediated by Transactivator Tas of Prototype Foamy Virus"

_viruses, 2023, doi:10.3390/v15091899_

Round 1

Reviewer 1 Report

Comments to the Author:

In this study, the authors analyzed the relation and interaction between DACH1 and PFV transactivator Tas. The author found that DACH1 mRNA levels were upregulated when cells were infected by PFV or transfected with Tas. However, no change was detected in protein expression level. The inconsistent transcriptional and protein levels were because of the SUMOylation of DACH1 induced by PPM1E.

This is a very interesting study. Experiments are well-designed and conducted. The manuscript is very straightforward and easy to read. It offers new perspectives on the function of DACH1 during virus infection.

Nevertheless, several concerns should be addressed. Detailed comments follow.

Major remarks:

1. The role or information of DACH1 in virology should be added in the introduction.

2. More background about Tas should be described in the introduction.

3. Figure 1. DACH1 mRNA expression was enhanced by Tas protein or Tas mRNA? Will DACH1 mRNA expression be upregulated when Hela cells were incubated with recombinant Tas protein?

4. Is Tas the only transactivator for PFV? Is there any other possible gene or protein of PFV, which may upregulate DACH1 mRNA expression?

5. Figure 2. Why Tas protein can bind both sequence of PFV and DACH1 promoter region? Is the binding site in the DACH1 promoter region similar to 5’-LTR and IP unique site of PFV? Could the author provide related sequence in the supplemental material?

6. Figure 2A. It will be better if the author can provide the sequences information of constructed promoter sequence deletion vector in supplemental material.

7. Why the author choose to mutate the sequence as shown in Figure 2B? What is the basis for such mutation strategy?

8. Line 270-271. “In vivo” is commonly used for experiment performed in animals or plants. Cell culture model is “in vitro” assay. The author should use suitable words for these two assays rather than “in vitro” and “in vivo”.

9. Figure 3. Did the author test the down-stream protein expression of DACH1? DACH1’s protein expression level was not influenced but the increased DACH1 mRNA level may influence other genes interacting with DACH1.

10. What is the role of PPM1E during the PFV infection?

11. It will be easier for the readers to understand the mechanism if the author can add a schematic diagram in the manuscript.

12. Line 445-447 “Comparison with DACH1-TRE……” please show the data in the supplemental material or reference.

13. Line 501-503. What cellular factors may be related to the inconsistent expression pattern of DACH1 transcription and translation level?

14. Figure 6. It will be make the data more solid if the author can provide the titer for progeny virus produced in DACH1 overexpressed cell lines.

15. In addition to SUMOylation, I am curious about the changes of other translational modification in DACH1 after FPV infection or Tas transfection?

16. It will be better if the author can polish the manuscript.

Minor remarks:

1. When the word like “DACH1” was first mentioned in the manuscript, the author should show its complete name. Same for PFA.

2. Figure 1B should indicate the meaning and differences of grey column and black column.

3. Reference for line 255. “there is a predicted sequence with high similarity to the PFV IP-TRE.”

4. Figure 2 “A\B\C\D” are in different sizes.

5. References for line 456-458.

6. line 52 “mammal cell” should be “mammalian cell”

7. Statistical Analysis should be mentioned in each figure.

8. line 502 typo “transcption”

It will be better if the author can polish the manuscript.

Reviewer 2 Report

The authors examine the role of DACH1 during PFV infection. They find that Tas binds to its promoter and stimulates RNA expression but perhaps surprisingly the protein levels are regulated by SUMOylation and degradation. This results in no net increase in protein levels. This is a well conducted study with writing that is easy to follow. While the reviewer doesn’t quite understand all the biology going on with upregulation and subsequent downregulation, this observation is valid and merits reporting. If the authors address the points below, I recommend for publication.

Minor points:

1- Could the authors add a figure (panel) showing the location of the vital G bases conserved within the promoters of DACH1-TRE sequences, Kip2-TRE, and IP-TRE?

2- Can the authors describe better how the PFV transductions were performed. In particular, I would like to know the PFV plasmids used to make the infectious virus.

3- As a follow up to their overexpression studies in Fig 6, it would be nice to see if the viral particles produced under overexpression of DACH1 are infectious. While it would be expected to have less viral particles, analysis of subsequent infectivity would indicate if there are additional effects that just decreased gag production.

Fine

Round 2

Reviewer 1 Report

The manuscript has addressed the issues that I raised before. I applaud the authors for their efforts to conduct this study and to improve their manuscript.